# The Predictive Power of Regression Models to Determine Grass Weed Infestations in Cereals Based on Drone Imagery—Statistical and Practical Aspects

Signe M. Jensen [1],*, Muhammad Javaid Akhter [2], Saiful Azim [1] and Jesper Rasmussen [1]

1 Department of Plant and Environmental Sciences, University of Copenhagen, Højbakkegaard Allé 13, 13626 Copenhagen, Denmark; saz@plen.ku.dk (S.A.); jer@plen.ku.dk (J.R.)
2 Research Centre Flakkebjerg, Department of Agroecology, Aarhus University, 8000 Aarhus, Denmark; javedakhter8864@gmail.com
* Correspondence: smj@plen.ku.dk

**Abstract:** Site-specific weed management (SSWM) may reduce herbicide use by identifying weed patches and weed-free areas. However, one major constraint is robust weed detection algorithms that are able to predict weed infestations outside of the training data. This study investigates the predictive power of regression models trained on drone imagery that are used within fields to predict infestations of annual grass weeds in the late growth stages of cereals. The main objective was to identify the optimum sampling strategy for training regression models based on aerial RGB images. The study showed that training based on sampling from the whole range of weed infestations or the extreme values in the field provided better prediction accuracy than random sampling. Prediction models based on vegetation indices (VIs) offered a useful alternative to a more complex random forest machine-learning algorithm. For binary decision-making, linear regression utilizing weed density information resulted in higher accuracy than a logistic regression approach that only relied on information regarding the presence/absence of weeds. Across six fields, the average balanced accuracy based on linear regression was in the range of 75–83%, with the highest accuracy found when the sampling was from the extreme values in the field, and with the lowest accuracy found for random sampling. For future work on training weed prediction models, choosing training sets covering the entire sample space is recommended in favor of random sampling.

**Keywords:** prediction models; validation; weed detection; weed monitoring; vegetation indices; precision agriculture; site-specific weed management (SSWM); UAV imagery

## 1. Introduction

Site-specific weed management (SSWM) is an environmentally friendly crop protection method that relies on technologies combining sensors, information systems, and informed management to optimize weed control and to reduce environmental impacts resulting from herbicide use [1,2]. The SSWM strategy deals with the fact that weeds are non-uniformly distributed in aggregated patches [3,4] and that herbicides can be saved when weed infestations are below the economical threshold. Hamouz et al. [5] tested different thresholds for different weed species and achieved herbicide savings in the range of 16–100% in winter wheat, depending on the herbicide and application threshold used. The achieved herbicide savings did not cause yield losses, showing the strong potentials of SSWM in arable farming.

Strategies for SSWM are either based on real-time detection and on-the-go spraying [6,7] or map-based methods where herbicides are targeted according to pre-loaded prescription maps [8,9]. Both methods rely on accurate weed monitoring and a variety of sensing devices, such as optoelectronic sensors. The use of RGB multispectral and hyperspectral cameras for weed detection have been thoroughly investigated [10,11].

Research on SSWM is mainly focused on developing new weed monitoring algorithms and procedures acknowledging that the robustness between different environments is a major constraint to successful SSWM implementation [12]. In a review of the potential use of sensor technology for weed detection, Peteinatos et al. [13] concluded that despite promising experimental results, major constraints to implementation remain since most of the significant results have been achieved under controlled conditions or following a specific protocol, which is difficult to replicate under normal field conditions. Moreover, the majority of the work on weed prediction models or algorithms has used subsets of the same fields on which the models have been trained for evaluating their performance. Even though such test data is non-overlapping with the training data, the test and training data cannot be considered independent. Accordingly, reported prediction accuracies will be overly optimistic at best and will often be downright useless for practical purposes. For a meaningful evaluation, completely new and independent fields should be the standard.

The combination of drones with cameras able to discern weeds [14,15] and GPS technologies providing geographical information for field mapping [16] makes the time-efficient monitoring of large areas possible.

The literature is full of agricultural applications of drones [17] and weed detection based on computer vision [2,11,18], but this study does not aim to contribute with cutting-edge knowledge on drone technology or machine learning. This work is based on tried and tested drone RGB imagery and the well-established knowledge that clearly visible weeds with different colors compared to the background have distinct spectral signatures. The vegetation indices (VIs) obtained from drone imagery are simple and effective algorithms that can be used to detect weeds. Two analytical approaches may be used. A qualitative approach where pixels are classified into weeds and a background or a quantitative approach where correlations between VIs and the weed density are utilized. The first approach is commonly used in drone weed mapping [19–21] and requires that only a limited number of pixels are mixed, which means that the image resolution has to match the size of the weeds or weed patches [22]. Therefore, segmentation algorithms work poorly if weeds or weed patches are smaller than the pixel size [23]. In such cases, quantitative approaches based on prediction models are relevant. Lambert et al. [24] showed that the density of *Alopecurus myosuroides* in wheat crops were correlated with spectral band combinations in RGB images. However, the results were not robust to an out-of-sample prediction from field-to-field, showing that weed monitoring became unreliable when the algorithms were used on images from fields that were not used during model parameterization.

Automated image analysis may work remarkably well when training and tests are performed on images from the same fields but fails when applied on images from new fields. This is also a problem for qualitative weed detection approaches that are based on segmentation. Therefore, Rasmussen et al. [21] realized that semi-automatic image analysis procedures including manual threshold adjustments were needed to establish the robust detection of *Cirsium arvense* in senescent cereals based on drone imagery.

Annual grasses constitute a major problem in winter cereals [25,26] and involve a substantial amount of herbicide use. This study takes the consequences of the lack of robust automated weed detection based on drone imagery and investigates the predictive power of regression models trained and used within fields to predict infestations of annual grass weeds in late growth stages, where weeds and crops are overlapping or occluding. The usual way of making prediction models for weed detection is to train it on a random subset of a field and then to test or apply the model on the remaining part of the field. To the authors' knowledge, no one has examined prediction accuracy effects using a strategic sampling strategy to find the subset of the field used to train the model. Therefore, the main objectives are to compare different strategies for training a prediction model and to compare different prediction strategies, i.e., aiming to predict the actual weed density or presence/absence of weeds. Grass weeds were detected in the late growth stages because this made it possible to take advantage of the color differences between the crop and the weeds (Figure 1). It is of practical relevance to map weeds in late growth stages when weed

patch distribution is relatively stable from one year to the next. This allows weed maps to be used for spraying the following year.

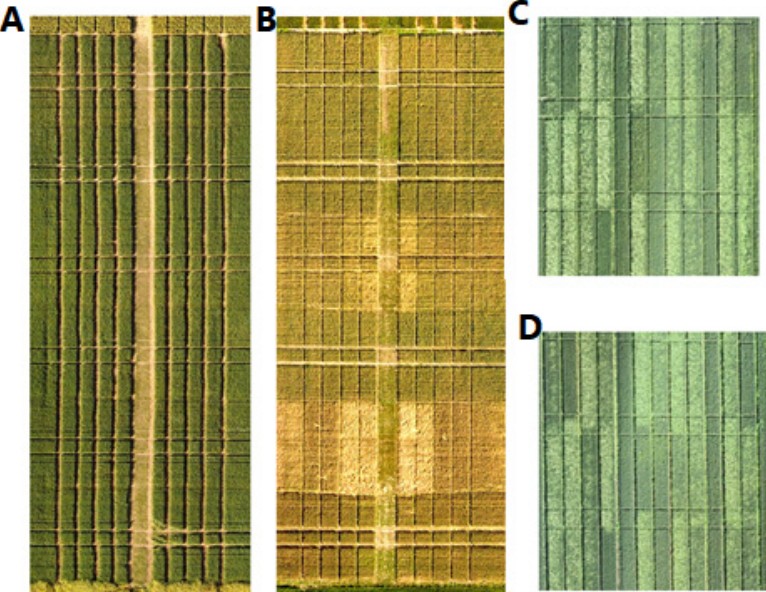

**Figure 1.** Orthomosaic from UAV imaging of selected fields to illustrate the visible plot variability due to weed infestations. (**A**): Field 1. (**B**): Field 5. (**C**): Field 6, early sowing. (**D**): Field 6, late sowing. Orthomosaics of the remaining fields are shown in Figure S1 in the Supplementary Materials.

The hypotheses guiding the present work were:

1.  Prediction models trained on extremes (plots with smallest and largest weed infestation) or plots across a whole range of weed densities provide better predictions than models trained on a random selection of plots;
2.  Increasing the number of plots used when training the model increases the prediction power;
3.  Increasing the variability in the weed infestation within fields increases the prediction power;
4.  Simple VI-based prediction models offer a useful alternative to more complex machine learning or artificial intelligence (AI)-based algorithms.

## 2. Materials and Methods

### 2.1. Experiments

Six fields were used to establish field-specific prediction models for annual grass weed infestations based on drone-derived VIs. The grass weeds were *Alopecurus myosuroides* (black-grass), *Vulpia myuros* (rattail fescue), and *Lolium multiflorum* (Italian ryegrass). Two spring barley fields were infested with *A. myosuroides* and *L. multiflorum*, respectively, one winter wheat field was infested with *A. myosuroides,* one was infested with *L. multiflorum*, and two winter wheat fields were infested with *V. myuros*. All of the fields were located in the eastern part of Denmark (Funen and Zealand) on sandy loam.

The selected study fields were chosen because there were large variations in the weed density, and the weed densities were recorded as a part of other experimental activities. Five fields were part of an integrated weed management demonstration project [27], and one field was part of a competition experiment [26]. Details about experimental treatments and results are not given in this article, as they are irrelevant to the prediction model issues raised in the present study.

In the integrated weed management demonstration project from 2017, each experiment consisted of 60 plots sized 2.5 m by 12 m. Five herbicide treatments including an untreated control and 12 tillage x sowing date combinations produced a wide range of grass densities

(Table 1). There were no block replications, and herbicide treatments varied according to weed species and herbicide treatment strategy. Details and experimental results are presented in Elander [27]. The weed competition experiment from 2019 was established with sown densities of V. myuros in winter wheat (Figure 1). It consisted of two main blocks representing early (13 September 2018) and late (3 October 2018) sown winter wheat. Within each sowing time, five V. myuros densities for two crop densities were established by varying the seeding rates. The seeding rates of V. myuros and winter wheat were adjusted according to sowing time. Three block replications produced a total of 60 plots that were 2.5 m by 10 m in size. Further details and results are presented in Akhter et al. [26].

**Table 1.** Distribution of grass weed densities. In Fields 1 to 4, density denotes flowering shoots in late growth stages, and in Fields 5 and 6, density denotes plants in spring.

| Field | Weed | Crop | Plots | Plots with Weeds (*n*) | Mean Density ($m^{-2}$) | Min. Density ($m^{-2}$) | Max. Density ($m^{-2}$) |
|-------|------|------|-------|------------------------|-------------------------|-------------------------|-------------------------|
| 1 | *Alopecurus myosuroides* | Spring barley | 60 | 32 | 1.4 | 0 | 12 |
| 2 | *Alopecurus myosuroides* | Winter wheat | 60 | 27 | 99.1 | 0 | 670 |
| 3 | *Lolium multiflorum* | Spring barley | 60 | 60 | 234.8 | 50 | 637.5 |
| 4 | *Lolium multiflorum* | Winter wheat | 60 | 55 | 477.4 | 0 | 2175 |
| 5 | *Vulpia myuros* | Winter wheat | 60 | 38 | 583.5 | 0 | 4437.5 |
| 6 | *Vulpia myuros* | Winter wheat | 60 | 47 | 117.6 | 0 | 484 |

*2.2. Crop and Weed Assessments*

In the integrated weed management demonstration project, the density of flowering grass shoots was counted in late June. In most plots, the density was counted in four randomly selected 0.25 m$^2$ quadrats per plot but for some plots, density assessments were based on 25 quadrants of 0.04 m$^2$. In plots with low densities of shoots, the total number of shoots per plot was counted. In the competition experiment, weeds were counted in April in two 0.25 m$^{-2}$ quadrates in each plot.

Aerial images were acquired with a Phantom 4 (DJI, Shenzhen, China) quadrotor fitted with an integrated RGB camera with a Sony 12.4 megapixel 1/2.3" CMOS sensor (4000 × 3000 pixels). The camera had a 20 mm (35 mm equivalent) focal length, a fixed f/2.8 aperture, and focus ∞. The camera was nadir-oriented. The flight altitude was 40 m, and the ground sampling distances (GSD) were 0.017 m pixel$^{-1}$. The flight-control app DJI GS Pro (DJI, Shenzhen, China, https://www.dji.com/dk/ground-station-pro, accessed on 1 June 2021) was used to generate efficient flight paths after setting the flight area, image overlap, and flight altitude. The frontal image overlap was 80%, and the lateral overlap was 70%. Images were captured on 3 July 2017 and 29 June 2019.

The orthomosaics were generated by the commercial photogrammetry software Pix4Dmapper (Pix4D, Lausanne, Switzerland, https://pix4d.com/ accessed on 1 June 2021). While processing in Pix4Dmapper, standard settings (Ag RGB template) were applied with a modification during the initial processing, where the calibration method chosen was "alternative" and where the internal parameter optimization chosen was "all prior" to ensure the best quality orthomosaics with the least spatial and spectral distortion possible. Image processing was conducted on drone images with ordinary GNSS (Phantom 4) and with the use of ground control points. However, the geometric performance of the orthomosaics was unimportant because the plots were manually cut based on visual on-screen evaluations. The open-source GIS software QGIS (https://www.qgis.org/en/site/ accessed on 1 June 2021) was used to cut plots and to calculate the mean values of the

digital numbers of red (R), green (G), and blue (B) reflectance bands for each plot in the produced orthomosaics.

Seven VIs based on the R, G, and B reflectance bands were considered for the prediction models; five regularly used indices, a single band (B), and luminosity (R + G + B):

Normalized Excess Green Index:

$$\mathrm{ExG} = \frac{2 \times \mathrm{G} - \mathrm{R} - \mathrm{B}}{\mathrm{R} + \mathrm{B} + \mathrm{G}} \tag{1}$$

Excess Green index—Excess Red index:

$$\mathrm{ExG} - \mathrm{ExR} = 2 \times \mathrm{G} - \mathrm{R} - \mathrm{B} - 1.4 \times \mathrm{R} + \mathrm{G} \tag{2}$$

Normalized difference index:

$$\mathrm{NDI} = (\mathrm{G} - \mathrm{R})/(\mathrm{G} + \mathrm{R}) \tag{3}$$

R B ratio:

$$\frac{\mathrm{R}}{\mathrm{B}} \tag{4}$$

Red index:

$$2 \times \mathrm{R} - \mathrm{G} - \mathrm{B} \tag{5}$$

Blue band:

$$\mathrm{Blue} = \mathrm{B} \tag{6}$$

R + G + B:

$$\mathrm{R} + \mathrm{G} + \mathrm{B} \tag{7}$$

The VIs were calculated for each plot based on the mean values of the R, G, and B reflectance bands. The latter two VIs were chosen because brightness seemed to be an indicator of weeds in a visual assessment of the orthomosaics (Figure 1, Figure S1). The VIs were used in the univariate prediction models as well as in multivariate models through the joint inclusion of one or more of the VIs.

The prediction models were trained to each field individually using different subsets of plots. Three different strategies for choosing the training plots were compared after dividing the plots into five groups: Group 1 contained the weed-free plots, while the rest of the plots were divided into four groups based on quantiles. The three strategies for choosing the training plots were:

*Random selection:* A random sample of plots (ignoring the groups);

*Selection from extremes*: A random sample of plots from the two groups with the fewest weeds (the weed-free plots and the plots with the lowest density of weed plants) combined with a random sample from the group of plots with the highest number of weed plants;

*Selection from the whole range*: A random sample from each of the five groups, ensuring that plots from the entire range of weed densities were included in the training set.

In addition, the different sizes of the training sets were evaluated. For a random selection, training sets of 10%, 20%, and 30% of all of the plots (a total of 6, 12, and 18 plots) were evaluated. For a selection from extremes, training sets of 10% and 20% (6 and 12 plots) were evaluated, while training sets of 1, 2, and 3 plots from each group (5, 10, and 15 plots) were evaluated to obtain a selection from the whole range.

### 2.3. Statistical Analysis

The analysis consisted of three steps. First, the fields were analyzed individually to determine the best description of the weed density according to the VIs and functional forms (step 1). Based on the results from step 1, the different strategies for selecting a training set to develop a prediction model were compared (step 2). Finally, a global model to cover all of the fields was developed for comparison with the within-field models.

The first step in the statistical analysis was within each field to determine the single most suitable VI and functional form describing the association between VI and weed density using linear regression models. The functional forms that were considered were linear, logarithmic, and exponential models. Comparisons were based on Akaike's Information Criteria (AIC) and $R^2$ based on all of the available data and root mean squared error (RMSE) based on leave-one-out cross-validation. AIC was used to compare the information lost compared to the underlying "true" but unknown model when a given statistical model was used to describe the grass weed density. $R^2$ was used to compare the variability in weed density that was accounted for by the different models. RMSE was used to compare the predictive performance. The comparisons based on these statistics were only relevant within a field, as they depended on the scale of the measurements and accordingly, onthe weed pressure. The leave-one-out cross-validation was conducted by fitting the model to all but one plot and evaluating the predicted value against the observed value for the remaining plot. This was completed by repeatedly leaving out a new plot each time, and the mean RMSE across all 60 plots was reported. The best VIs and functional form were found by combining and comparing the information from AIC, $R^2$, and RMSE.

In the second step of the analysis, the most promising VI and functional form for each field was used to evaluate different strategies for selecting a training set for the prediction model. For each sampling strategy, a thousand training sets were generated by random sampling. For each training set, the performance of the prediction model was validated on the remaining part of the data set. The performance of the different strategies was compared using $R^2$ and RMSE.

The strategies for sampling a training set were also evaluated for binary decision making, where the ability to predict the presence or absence of weeds was validated in terms of sensitivity (how many of the weed plots were detected) and specificity (how many of the weed-free plots were detected). In addition, for easier comparison to other studies, the performance of the sampling strategies was summarized in terms of balanced accuracy, which can be defined as:

$$\text{Balanced accuracy} = \frac{\text{sensitivity} + \text{specificity}}{2} \tag{8}$$

The prediction of the presence/absence of weeds was based on the linear regressions by dichotomizing the weed density predictions. Furthermore, to assess a training approach that demanded less work, the presence/absence of weeds was used to train a binary prediction model that was based on logistic regression. For both approaches, the threshold used for dichotomizing the predictions were based on optimizing Youden's index [28]:

$$\text{Youden's index} = \text{sensitivity} + \text{specificity} - 1 \tag{9}$$

The third step in the analysis of the grass weed data was to find a global model for the prediction of weeds. Two types of models were considered, multiple linear regression and random forest regression. In both types of models, a global model was estimated to the joint data set of five of the six fields, with the remaining field being used as test data. The performance of the prediction model was assessed in terms of the RMSE of the predictions in the test set. This procedure was repeated with each of the six fields that were left out of the model training and that instead acted as test data. In multiple linear regression, the number and selection of predictor variables to include were based on 10-fold cross-validation with RMSE as criteria, i.e., the training set was split randomly into ten subsets, with a model fitted to the combination of nine of the subsets and the last being used for evaluating the model performance. This was repeated ten times by rotating the subset to be left out during model fitting [29]. The inclusion of 1–9 VIs and single bands, giving a total of 452 distinct models, were evaluated in the cross-validation. To avoid the number of models to be compared in the cross-validation from increasing further, the VIs and single bands were only included in their linear form. In the random forest regression, the number of variables that was sampled for splitting at each node was set to 4 (found by

cross-validation), the minimum size of the terminal nodes was set to 5, and 500 trees were used to develop the model.

All of the analyses were made in the open-source statistical programming software R version 4.0.2 [30]. In particular, the extension package randomForest was used [31].

## 3. Results

The within-field evaluation of the association between different VIs and grass weed density using different functional forms showed similar conclusions using AIC, predicted RMSE, and $R^2$. Generally, no single VI performed well across all fields.

Figure 2 presents the $R^2$ for the different fields and VIs. In Fields 1, 3, and 6, the blue band performed the best out of all of the VIs that were considered. For Field 6 in particular, R + G + B was a close competitor, while R/B was a close competitor to the blue band for Field 3. In Field 1, the blue band only performed slightly better than the rest of the VIs. In Field 2, the grass weed density was best described using R/B, while the density in Field 4 was best described by the red index. In Field 5, R + G + B was the best performing index. From the functional forms that were considered, the linear form generally resulted in the best or close to the best performance. The same conclusions for VIs and functional forms were reached through model comparisons using AIC and RMSE, with a few exceptions: In Field 1, RMSE found ExG to be the best VI, but it was closely followed by the blue band. In Field 3, AIC found R/B to be the best VI, but it was closely followed by the blue band. In Field 6, RMSE found R + G + B to be the best performing VI, but it was closely followed by the blue band. The RMSE in Field 1 was generally low for all of the VIs, as the weed pressure was low. The results for AIC and RMSE are presented in Tables S1 and S2 in the Supplementary Materials.

Figure 3 shows the prediction models based on the linear form and the best performing VI in each of the six fields individually. The prediction RMSE differed largely between the different fields. The lowest prediction RMSE for Field 1 was 1.6, while the lowest observed RMSE for Fields 2–6 was 91.5, 95.2, 311.6, 449.1, and 80.1, respectively.

For each of the strategies for selecting a training set, the ability of the resulting model to predict the association between VI and weed density is presented in Tables 2 and 3 in terms of prediction the $R^2$ and RMSE in the test set, respectively. Similar results for the training sets are reported in Tables S3 and S4 in the Supplementary Materials. No strategy for selecting the training set consistently produced better results. In terms of $R^2$, training a prediction model based on selection from the whole range and based on a random selection of plots performed similarly, whereas a lower $R^2$ was found when the prediction model was trained by random selection from the extremes. The highest $R^2$ values were reached in the fields with a large range of weed densities, while the lowest $R^2$ values were found for Field 1, which had a narrow range of weed densities. In terms of RMSE, the strategy of selecting the training set from the whole range of densities generally resulted in the lowest RMSE, with one clear exception being Field 5, where the lowest RMSE values were obtained by selecting from the extremes. Due to the low weed density in Field 1, a low RMSE but also a low $R^2$ was observed for all of the strategies.

Figure 4 shows the development in $R^2$ and RMSE for the test data with an increasing percentage of the data set being randomly sampled as training data. For all of the fields, with the exception of Field 5, an increase in the size of the training set was associated with a slight increase in the prediction accuracy. The $R^2$ values were almost constant for the training sets that were composed of 10% to 50% of the data set. Similarly, no real changes in the RMSE could be seen for the training sets consisting of 20% to 90% of the data set. For Field 5, a decreasing $R^2$ but also a decreasing RMSE was observed when the training set increased in size.

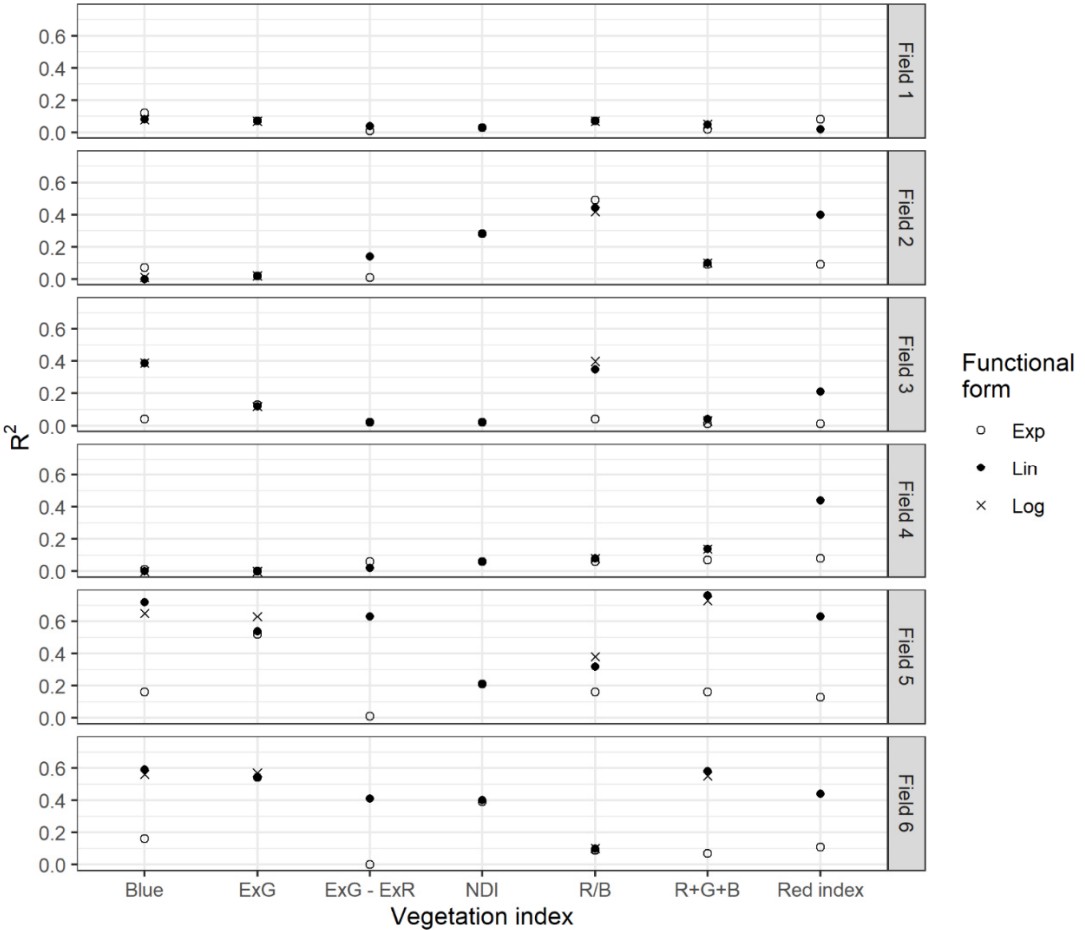

**Figure 2.** $R^2$ for the prediction models based on all data for each field for different vegetation indices and different functional forms of the vegetation index.

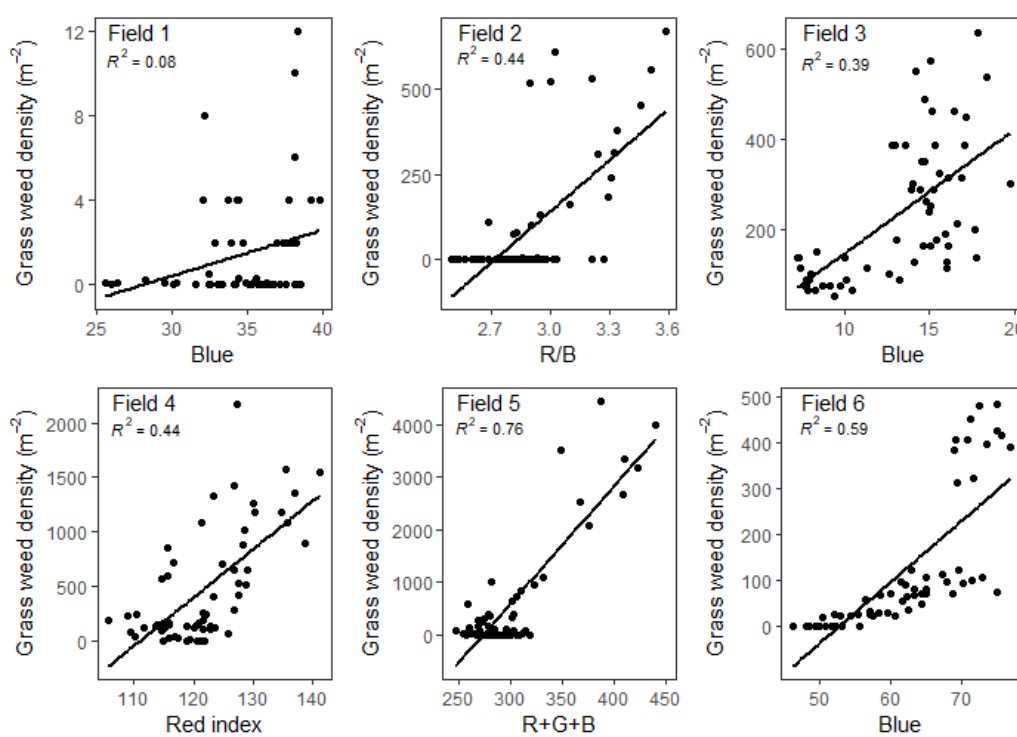

**Figure 3.** Predictions for each field based on the best-performing vegetation index.

**Table 2.** Prediction $R^2$ of different strategies by selecting a subset of data to train the linear prediction model. Results are given for different percentages (number of plots) used as training sets. For the small training sets of 8% or 10%, the highest $R^2$ values within each field are presented in bold.

| Field | VI | Random Selection | | | Selection from Extremes | | Selection from the Whole Range | | |
|---|---|---|---|---|---|---|---|---|---|
| | | **10% (6)** | **20% (12)** | **30% (18)** | **10% (6)** | **20% (12)** | **8% (5)** | **16% (10)** | **25% (15)** |
| 1 | Blue | **0.08** | 0.08 | 0.08 | **0.08** | 0.08 | **0.08** | 0.07 | 0.07 |
| 2 | R/B | **0.44** | 0.44 | 0.44 | 0.43 | 0.39 | 0.43 | 0.42 | 0.39 |
| 3 | Blue | **0.39** | 0.39 | 0.40 | 0.36 | 0.33 | 0.38 | 0.37 | 0.36 |
| 4 | Red index | 0.44 | 0.45 | 0.45 | 0.43 | 0.39 | **0.45** | 0.45 | 0.45 |
| 5 | R +G + B | 0.76 | 0.76 | 0.76 | 0.74 | 0.68 | **0.77** | 0.76 | 0.77 |
| 6 | Blue | **0.59** | 0.59 | 0.59 | 0.55 | 0.50 | **0.59** | 0.59 | 0.59 |

**Table 3.** Average prediction RMSE of different strategies for selecting a subset of data to train the prediction model. Results are given for different percentages (number of plots) used as training sets. For the small training sets of 8% or 10%, the lowest RMSE values within each field are presented in bold.

| Field | VI | Random Selection | | | Selection from Extremes | | Selection from the Whole Range | | |
|---|---|---|---|---|---|---|---|---|---|
| | | **10% (6)** | **20% (12)** | **30% (18)** | **10% (6)** | **20% (12)** | **8% (5)** | **16% (10)** | **25% (15)** |
| 1 | Blue | **3.0** | 2.7 | 2.6 | 4.7 | 2.9 | 3.6 | 3.0 | 2.8 |
| 2 | R/B | 178.3 | 156.1 | 149.5 | 197.0 | 147.8 | **164.5** | 149.5 | 144.6 |
| 3 | Blue | 146.4 | 129.4 | 125.4 | 181.1 | 179.9 | **132.0** | 126.7 | 126.0 |
| 4 | Red index | 475.8 | 423.9 | 406.4 | 473.5 | 415.0 | **459.9** | 414.8 | 407.3 |
| 5 | R + G + B | 837.7 | 681.4 | 619.9 | **623.6** | 540.0 | 736.6 | 617.2 | 583.2 |
| 6 | Blue | 121.5 | 108.6 | 104.6 | 137.3 | 139.6 | **104.9** | 100.5 | 99.0 |

Figure 5 shows the distribution of the predicted $R^2$ from the 1000 random samples from each selection strategy and each field. Generally, random selection and selection from extremes resulted in a larger range of $R^2$ values, whereas the 1000 samples found by randomly sampling from the entire range of densities were gathered more closely around the mean, with fewer extreme performances.

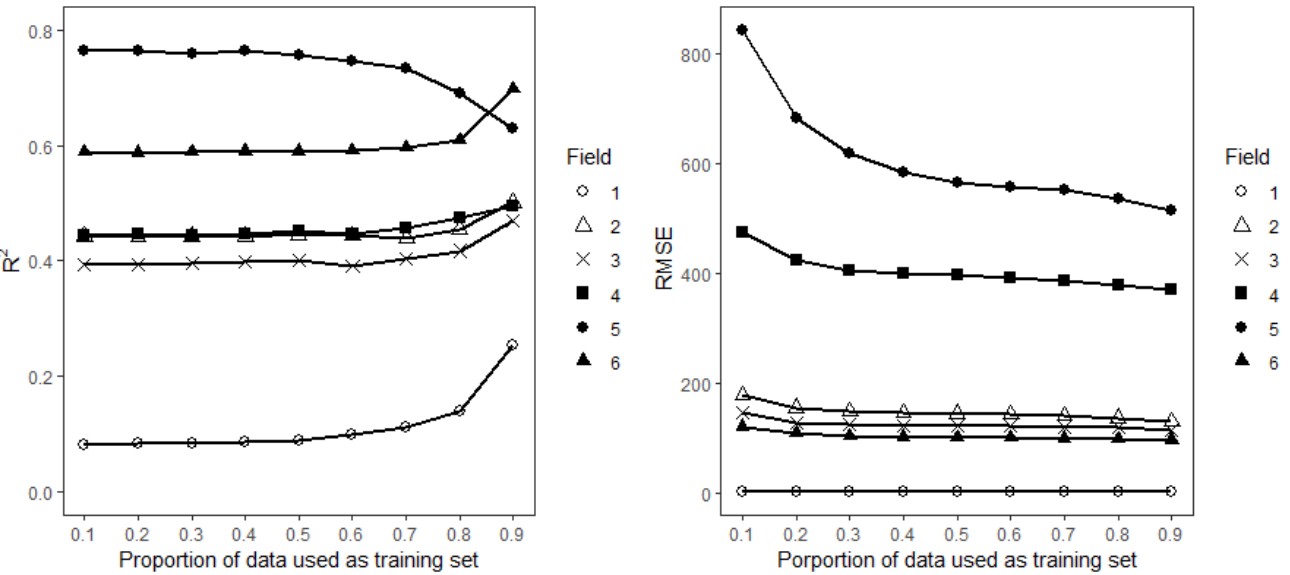

**Figure 4.** Development in $R^2$ and RMSE based on predictions with an increasing size of the training set when the training set was chosen by random sampling.

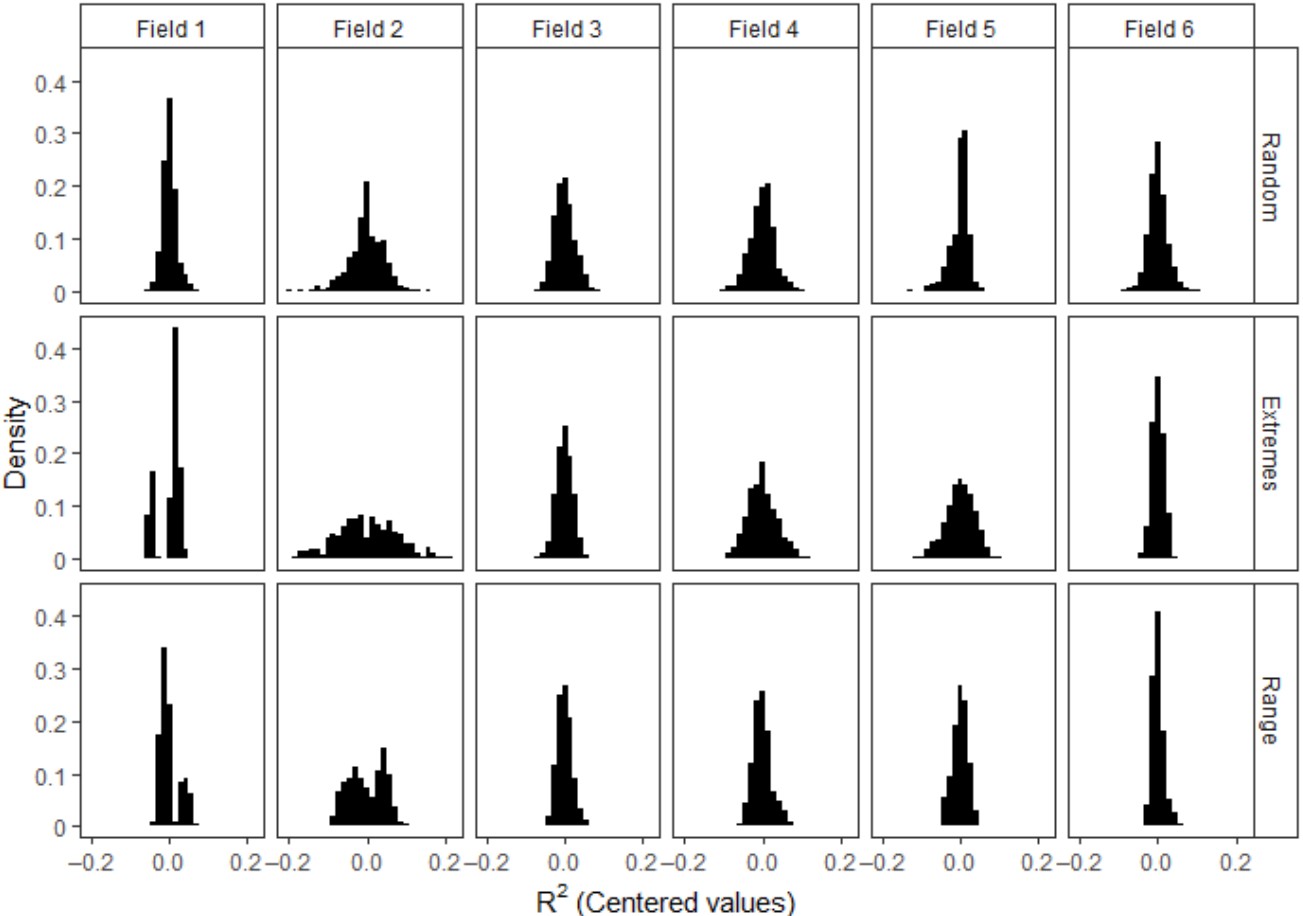

**Figure 5.** Distribution of predicted $R^2$ values found by random sampling plots according to three different strategies. For random sampling and sampling from extremes, 10% (six) of the plots were used. For sampling from the entire range, 8% (five) of the plots, one from each group, were used. The histograms are based on 1000 repetitions of each sampling strategy within each field. The histograms are centered around the mean within the field and selection strategy.

From an SSWM perspective, it is relevant to know whether weeds are present or not. Table 4 presents sensitivity, specificity, and balanced accuracy when the presence of weeds was predicted from linear regression models. Table 5 presents similar results for models where training the model only implied assessing the presence/absence of weeds. Results on the performance of the prediction models in the training set are presented in the Supplementary Materials (Tables S5 and S6).

In Field 3, where all of the plots contained weeds, the logistic regression approach resulted in a sensitivity of 1 for all of the strategies, while the linear regression approach resulted in a sensitivity ranging from 0.95 to 1. Selecting the training set from the extremes generally resulted in the highest sensitivity and specificity for the linear regression approach (Table 4). For the logistic regression approach (Table 5), selection from the whole range resulted in the highest sensitivity, while selecting from the extremes resulted in the highest specificity. The balanced accuracy across all of the fields only revealed minor overall differences between the selection strategies for the logistic regression approach, with a small advantage being provided when selecting from the whole range. For the linear regression approach, selecting from the extremes resulted in the highest balanced accuracy, with a considerable gain being incurred compared to when the model was trained on a random selection. Increasing the size of the training set did not necessarily result in higher sensitivity or specificity.

**Table 4.** Sensitivity, specificity, and balanced accuracy for different strategies for selecting a subset of data to train the linear prediction model to describe the association between vegetation index and grass weed density. Results are given for different percentages (number of plots) used as training sets. For the small training sets of 8% or 10%, the highest sensitivity and specificity within each field are presented in bold.

| | Field | VI | Random Selection | | | Selection from Extremes | | Selection from the Whole Range | | |
| --- | --- | --- | --- | --- | --- | --- | --- | --- | --- | --- |
| | | | 10% (6) | 20% (12) | 30% (18) | 10% (6) | 20% (12) | 8% (5) | 16% (10) | 25% (15) |
| Sensitivity | | | | | | | | | | |
| | 1 | Blue | 0.52 | 0.60 | 0.63 | **0.73** | 0.61 | 0.57 | 0.62 | 0.61 |
| | 2 | R/B | 0.67 | 0.80 | 0.83 | **0.84** | 0.78 | 0.81 | 0.87 | 0.87 |
| | 3 | Blue | **0.98** | 0.99 | 1 | 0.95 | 0.98 | **0.98** | 0.99 | 1 |
| | 4 | Red index | **0.85** | 0.86 | 0.85 | 0.80 | 0.77 | 0.75 | 0.79 | 0.80 |
| | 5 | R + G + B | 0.66 | 0.77 | 0.83 | **0.83** | 0.89 | 0.77 | 0.83 | 0.84 |
| | 6 | Blue | 0.93 | 0.93 | 0.94 | **0.95** | 0.96 | **0.95** | 0.94 | 0.94 |
| Specificity | | | | | | | | | | |
| | 1 | Blue | 0.62 | 0.61 | 0.63 | **0.73** | 0.63 | 0.64 | 0.59 | 0.57 |
| | 2 | R/B | 0.75 | 0.73 | 0.73 | **0.78** | 0.85 | 0.69 | 0.70 | 0.70 |
| | 3 | Blue | - | - | - | - | - | - | - | - |
| | 4 | Red index | 0.60 | 0.63 | 0.64 | **0.74** | 0.74 | 0.72 | 0.73 | 0.72 |
| | 5 | R + B + G | 0.73 | 0.75 | 0.78 | **0.93** | 0.94 | 0.72 | 0.70 | 0.70 |
| | 6 | Blue | **0.72** | 0.71 | 0.70 | 0.65 | 0.52 | **0.72** | 0.74 | 0.75 |
| Accuracy | | | | | | | | | | |
| | All | | 0.75 | 0.78 | 0.80 | **0.83** | 0.81 | 0.78 | 0.79 | 0.79 |

**Table 5.** Sensitivity, specificity, and balanced accuracy for different strategies for selecting a subset of data to train the prediction model using a logistic regression model to describe the association between vegetation index and probability of grass weed density. Results are given for different percentages (number of plots) used as training sets. For the small training sets of 8% or 10%, the highest sensitivity and specificity within each field are presented in bold.

| | Field | VI | Random Selection | | | Selection from Extremes | | Selection from the Whole Range | | |
| --- | --- | --- | --- | --- | --- | --- | --- | --- | --- | --- |
| | | | 10% (6) | 20% (12) | 30% (18) | 10% (6) | 20% (12) | 8% (5) | 16% (10) | 25% (15) |
| Sensitivity | | | | | | | | | | |
| | 1 | Blue | 0.39 | 0.39 | 0.38 | 0.33 | 0.37 | **0.47** | 0.44 | 0.43 |
| | 2 | R/B | 0.47 | 0.55 | 0.57 | 0.47 | 0.36 | **0.60** | 0.60 | 0.59 |
| | 3 | Blue | **1** | 1 | 1 | **1** | 1 | **1** | 1 | 1 |
| | 4 | Red index | **0.75** | 0.65 | 0.59 | 0.62 | 0.47 | 0.43 | 0.48 | 0.47 |
| | 5 | R + G + B | 0.43 | 0.45 | 0.46 | 0.25 | 0.17 | **0.49** | 0.52 | 0.51 |
| | 6 | Blue | **0.83** | 0.87 | 0.88 | 0.75 | 0.84 | 0.82 | 0.88 | 0.88 |
| Specificity | | | | | | | | | | |
| | 1 | Blue | 0.61 | 0.60 | 0.61 | **0.66** | 0.62 | 0.51 | 0.51 | 0.54 |
| | 2 | R/B | 0.72 | 0.71 | 0.71 | **0.79** | 0.85 | 0.61 | 0.64 | 0.67 |
| | 3 | Blue | - | - | - | - | - | - | - | - |
| | 4 | Red index | 0.24 | 0.32 | 0.38 | 0.43 | 0.61 | **0.66** | 0.68 | 0.69 |
| | 5 | R + B + G | 0.65 | 0.69 | 0.69 | **0.88** | 0.93 | 0.61 | 0.63 | 0.65 |
| | 6 | Blue | 0.72 | 0.86 | 0.89 | 0.83 | 0.88 | **0.93** | 0.91 | 0.90 |
| Accuracy | | | | | | | | | | |
| | All | | 0.65 | 0.67 | 0.68 | 0.67 | 0.68 | **0.68** | 0.69 | 0.69 |

The joint prediction model trained on the data set consisting of all of the data from five fields and evaluated on the remaining field resulted in a prediction RMSE of 196.7, 394.3, 2271.3, 471.4, 853.3, and 501.8 using the multiple linear regression approach, and a prediction RMSE of 104.5, 302.7, 198.7, 518.0, 1063.0, and 118.0 using the random forest regression. Figures 6 and 7 show the observed vs. predicted weed densities based on the random forest regression in the training data and test data, respectively, for each field used as test data. In all of the scenarios, the prediction model described the densities in the training set reasonably well (Figure 6), but in most of the test sets, the predictions were far from the observed values (Figure 7).

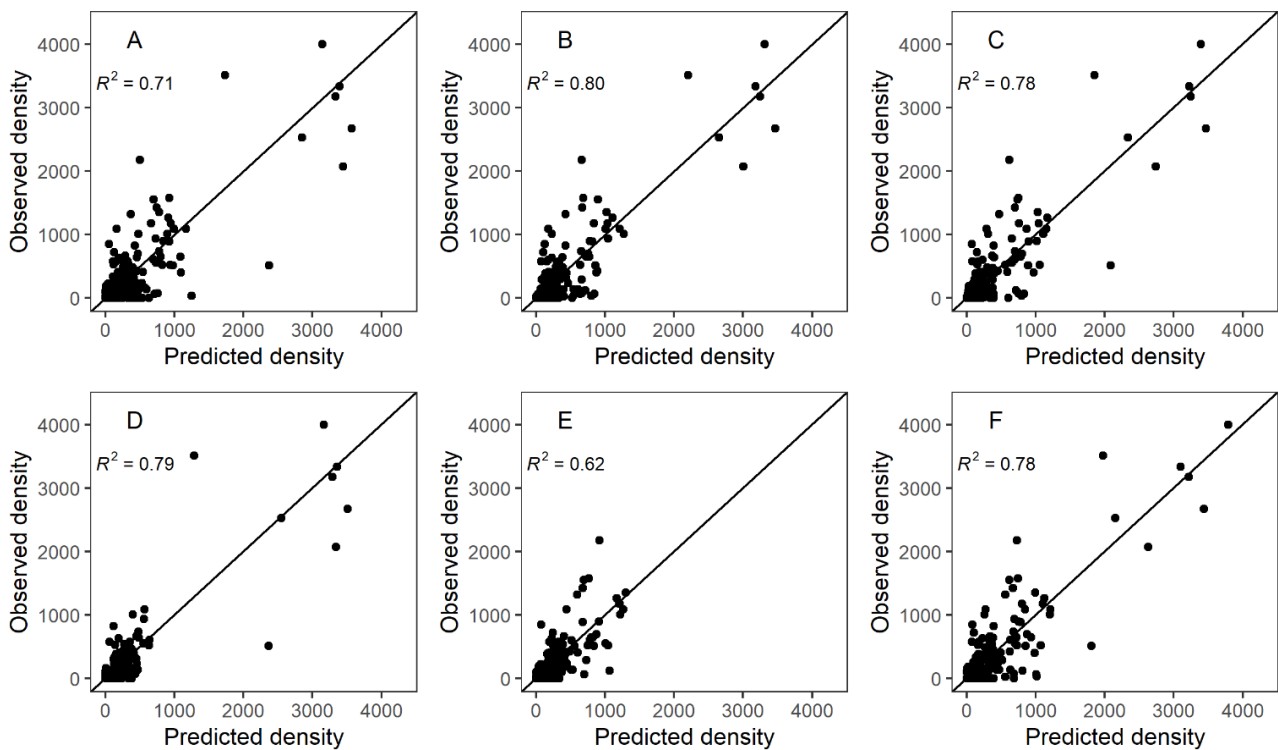

**Figure 6.** Observed vs. predicted weed density in training data. Five fields are used as training data set, and the remaining field is used as test data in a random forest regression. (**A**): Field 1 is used as test data. (**B**): Field 2 is used as test data. (**C**): Field 3 is used as test data. (**D**): Field 4 is used as test data. (**E**): Field 5 is used as test data. (**F**): Field 6 is used as test data.

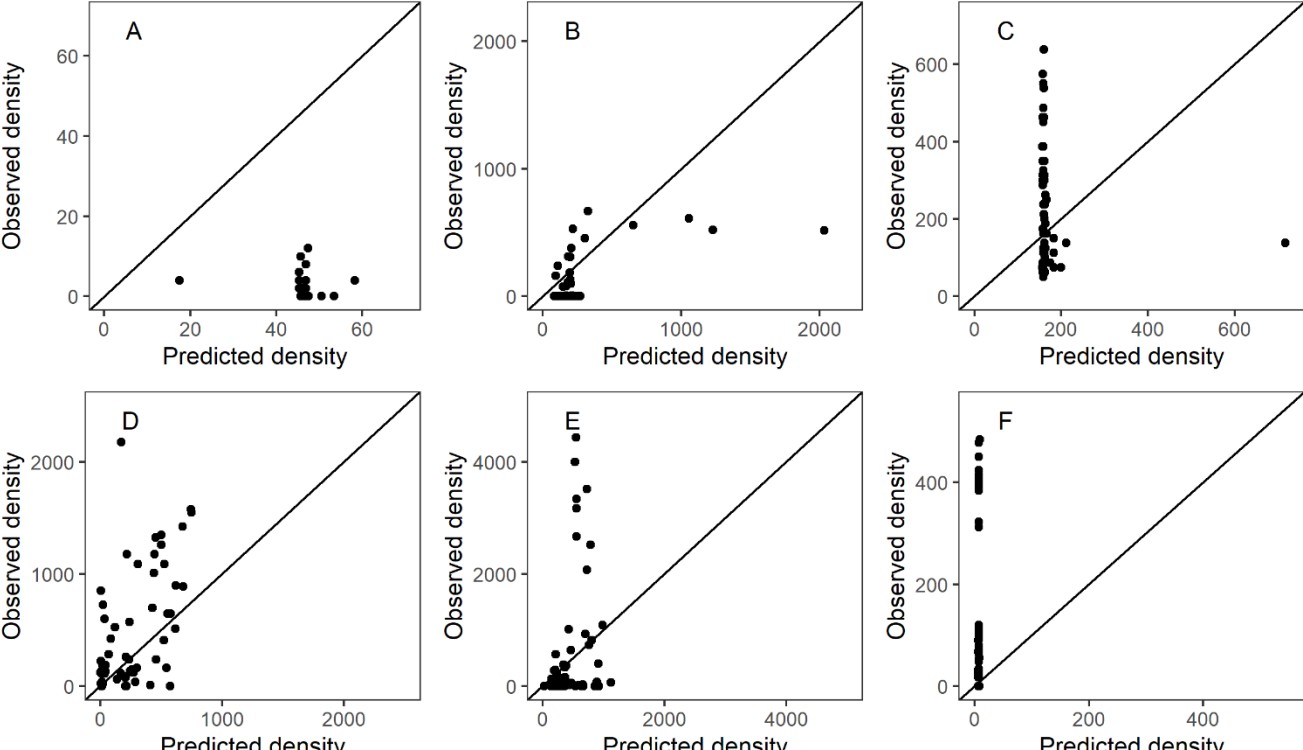

**Figure 7.** Observed vs. predicted weed density in independent test data. Five fields are used as training data set and the remaining field is used as test data in a random forest regression. (**A**): Field 1 is used as test data. (**B**): Field 2 is used as test data. (**C**): Field 3 is used as test data. (**D**): Field 4 is used as test data. (**E**): Field 5 is used as test data. (**F**): Field 6 is used as test data.

## 4. Discussion

Weed detection is an area that is in rapid development and that has been taking advantage of the developments in machine learning (ML) and deep learning (DL) algorithms [11,12]. The present work focused on a small area in weed detection: selecting a proper training set to optimize the prediction accuracy. In general, it was found that choosing a random sample to compose the training set resulted in lower prediction accuracy than when selecting from the extreme values or when selecting from the whole range of densities. Accuracies were found to be higher when the full knowledge of the densities was included rather than the presence/absence of weeds. Here, the average balanced accuracy increased from 75% to 83% when the training set was chosen from the extremes compared to choosing the training set as a random sample, while the average RMSE was reduced by 4.4% when choosing the training set comprising the whole range of densities compared to choosing the training set comprising a random sample.

In the present work, the joint prediction models for pre-harvest weed infestations were tested on independent fields to provide a realistic evaluation of their performance in practice. We found that it was not possible to fit a global/joint prediction model that was robust to new fields, and in several cases, the results from the joint models were meaningless predictions. In comparison, for all of the fields, the estimated prediction RMSE values from the models trained within the field were lower, even if only 10% of the plots were used to train the model. The joint models that were considered here were based on multiple linear regression and random forest regression for training the model on the joint data set. Without a doubt, a better model could be developed from more complex machine learning (ML) or deep learning (DL) algorithms by taking advantage of spectral features, spatial patterns, biological morphology, and visual textures [32,33]. However, in scenarios with a high degree of leaf overlap, as is the case with grass weed in cereals, the prediction performance is lower [34]. The robustness of a trained algorithm to a new environment is further complicated due to differences in soil type and conditions, plant stage, and the color and morphology of both the weeds and the crop due to cultivar differences [12]. Further, when using UAV imaging, the changes in the type and resolution of the camera, light conditions, and flight altitude will likely also influence the performance [35,36].

Due to trade-offs between precision, cost-effectiveness, and the availability of more sophisticated procedures to farmers, relatively crude weed mapping procedures, such as the ones examined here, may be more feasible [12]. The main challenge for SSWM is to develop and implement weed monitoring systems that are practically feasible in current farming systems. Monitoring systems do not have to be perfect. If the effort put into training the model exceeds the gain from the predictions, then it makes no sense to employ the technology. Therefore, to make it meaningful to train a prediction model within the field, the training set needs to be rather small. This adds constraints to the type of models that can be used, as models with low degrees of freedom, i.e., those containing a large number of variables relative to the number of observations, will usually suffer from overfitting and an associated low prediction accuracy outside of the training set. This is the case for both multiple regression and ML/AI algorithms [29].

In the present work, different assessment criteria were used to evaluate the predictive performance of the models. The reason for this was that thedifferent assessment criteria have different purposes and may produce different conclusions. Simply assessing the associations may not reveal the best prediction ability, as was also seen in the results for some of the fields and selection strategies [29].

When assessing the different strategies for selecting the training set in terms of predicting the weed density, the mean $R^2$ values were similar for the random selection and the selection from the whole range strategy, and lower for the strategy of selecting from the extremes. However, the distribution of the possible $R^2$ values was closer to the mean for the strategy of selecting from the whole range. Accordingly, even though the two strategies on average performed similarly (in terms of $R^2$), the risk of obtaining a much worse performance was lower when following the strategy of sampling from the whole

range. In other words, the probability that the sampling strategy (due to the random sampling involved) results in a much worse prediction model is lower when there is a guarantee that the whole range of densities is covered. In terms of RMSE, selecting from the whole range was preferred.

The performance of the sampling strategy for binary decision-making was assessed for threshold optimization based on Youden's index that weights sensitivity and specificity equally [37,38]. Other optimization criteria for selecting threshold values could be just as relevant depending on the focus of the prediction, e.g., the area under the curve (AUC) for the receiver operating characteristic (ROC) curve, a partial ROC AUC, or sensitivity for a fixed specificity [39]. Generally, increasing the threshold will decrease the number of predicted plots with weeds, decreasing the number of false positives but also decreasing the number of true positives. Decreasing the threshold will increase the number of predicted plots with weeds, decreasing the number of false negatives but also decreasing the number of true negatives. One could argue that in weed detection, it is most important to be able to identify the areas of the field containing weeds, i.e., maximizing the sensitivity. However, the true negative findings (the specificity) relate to the areas where spraying can be safely avoided, and accordingly, where more savings can be achieved compared to the alternative of spraying the whole field.

A general conclusion of an increased prediction power with an increased size of the training set could not be reached. For most selection strategies, an almost constant or sometimes even decreasing performance was found. Increasing the size of the training set generally resulted in only slightly higher precision in terms of RMSE (Table 3, Figure 5), but no real change in $R^2$ (Table 2, Figure 5). When precision was evaluated for binary decision making, the specificity (ability to detect the weed-free plots) was found to decrease with an increasing size of the training set for several of the selection strategies and fields (Tables 4 and 5). This indicated that the risk of over-fitting the model to the training data was real for too large training sets.

When assessing the influence of an increasing proportion of the data set used for training the model, Field 5 showed a decreasing $R^2$ but also a decreasing RMSE when the size of the training set increased. This phenomenon may be explained by a bias–variance trade-off. An increasing model complexity will typically decrease bias but will increase the variance [29]. The optimal model complexity will depend on several aspects, including the type of model and the data set at hand, which may also explain why the decreasing $R^2$ is only observed for a single field.

We hypothesized that the prediction power was higher for fields with high variability in the weed pressure, implying more information on the association between VI and weed density. This was partly supported in the association analyses where the $R^2$ for the predictions for Field 1 with a very low weed pressure was very low compared to the other fields, no matter the choice of strategy for selecting the training set. Similarly, the highest prediction $R^2$ was obtained for Field 5, which was also the field with the largest span in weed densities. However, moving to the results based on a binary decision, a larger variation does not necessarily seem to be an advantage. For both the linear and generalized linear approaches, the best prediction accuracies in terms of both sensitivity and specificity were achieved for Field 6. Another important aspect of variability may be the general field variation relative to the variation explained by the weed. Variation other than the one in focus may complicate the weed density predictions. The inclusion of pedo-environmental and climatic information in combination with phenological and spectral characteristics could possibly improve a prediction model for such a setting. The data used in the present work are from relatively small and homogenous fields compared to SSWM scenarios where expectedly, the fields will be bigger, and accordingly, they may be more heterogeneous, not only in terms of weeds but also in terms of other factors (e.g., crop growth stage and soil conditions). Future work should investigate how the general variation in crop growth conditions in large fields influences the performance of prediction models.

The focus of the present work was on the statistical considerations for developing weed prediction models. Therefore, we also believe that these considerations apply to a larger range of crop and weed species besides those used for illustration in the present work.

## 5. Conclusions

For grass weed detection in cereals, simple VI-based prediction models trained within the field were shown to offer a useful alternative to more complex joint machine learning or AI-based algorithms aiming to cover all fields. The use of independent test fields showed that predictions outside of the training data gave unreliable predictions. It was shown that an increasing number of plots used in training does not necessarily increase the prediction power for within-field prediction. In addition, if the goal is binary decision-making, the extra information obtained by knowing the weed density compared to assessing presence/absence of weeds increases the prediction accuracy. Compared to a random selection of plots for training, a strategic selection ensuring that the whole range or even just extreme values are represented may result in better prediction accuracy. Therefore, it is suggested that for future work on training prediction models, the use of more strategically chosen training sets covering the sample space should be considered in favor of random sampling.

**Supplementary Materials:** The following are available online at https://www.mdpi.com/article/10.3390/agronomy11112277/s1, Table S1: AIC differences (AIC-min AIC) for the prediction models based on all data for each field for different vegetation indices and different functional form of the vegetation index. Table S2: RMSE for the prediction models based on all data for each field for different vegetation indices and different functional form of the vegetation index. Table S3: Average $R^2$ for training sets of different strategies for selecting a subset of data to train the linear prediction model. Table S4: Average RMSE for the training sets of different strategies for selecting a subset of data to train the prediction model. Table S5: Sensitivity, specificity, and balanced accuracy in the training sets for different strategies for selecting a subset of data to train the linear prediction model to describe the association between vegetation index and grass weed density. Table S6: Sensitivity, specificity, and balanced accuracy in the training sets for different strategies for selecting a subset of data to train the prediction model using a logistic regression model to describe the association between vegetation index and probability of grass weed density. Figure S1. Orthomosaic from UAV imaging of selected fields to illustrate the visible plot variability due to weed infestations.

**Author Contributions:** Conceptualization, S.M.J. and J.R.; data curation, M.J.A., S.A. and J.R.; formal analysis, S.M.J.; methodology, S.M.J. and J.R.; visualization, S.M.J.; writing—original draft, S.M.J. and Jesper Rasmussen; writing—review and editing, S.M.J., M.J.A., S.A. and J.R. All authors have read and agreed to the published version of the manuscript.

**Funding:** The study was conducted as a part of the projects Future Cropping (J.nr. 5107-00002B), Innovation Fund Denmark.

**Data Availability Statement:** The data presented in this study are openly available in the Electronic Research Data Archive University of Copenhagen at https://doi.org/10.17894/ucph.7b4f3336-050f-4810-9410-41344da3767a.

**Conflicts of Interest:** The authors declare no conflict of interest.

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
