# Peer review of "The Predictive Power of Regression Models to Determine Grass Weed Infestations in Cereals Based on Drone Imagery—Statistical and Practical Aspects"

_agronomy, doi:10.3390/agronomy11112277_

Round 1
Reviewer 1 Report
No additional revisions are needed.
The authors correctly addressed all the highlighted issues.
Author Response
Thank you
Reviewer 2 Report
I have now finished reading manuscript. It is a well written paper.
I have some minor comments appended as under;
My major question is why authors used late growth stage to train the regression models? Weeds are abundant at the earlier growth phase of wheat and rarely controlled at later stages. Should not training at earlier growth stage had more promising results?
Please change Site-specific weed management (SSWM) may optimize weed control and reduce herbicide use by identifying weed patches and weed-free areas to -> Site-specific weed management (SSWM) may reduce herbicide use by identifying weed patches and weed-free areas.
VI-based - Please elaborate at the first cite
There is no concrete conclusion at the end of the abstract section. What the study suggests for the future studies?
Line 111. The grass weeds were A. myosuroides, please write full form at the first cite of the weed name in each section and then abbreviate the genus name
Table 1. The names of the weed species should be written in full
The rest of the sections are ok.
Author Response
My major question is why authors used late growth stage to train the regression models? Weeds are abundant at the earlier growth phase of wheat and rarely controlled at later stages. Should not training at earlier growth stage had more promising results?
We have added the following to the introduction: “The grass weeds were detected in late growth stages because this made it possible to take advantage of color difference between crop and weeds (Figure 1). It is of practical relevance to map weeds in late growth stages when weed patch distribution is relatively stable from one year to the next. This allows weed maps to be used for spraying the following year.”
Please change Site-specific weed management (SSWM) may optimize weed control and reduce herbicide use by identifying weed patches and weed-free areas to -> Site-specific weed management (SSWM) may reduce herbicide use by identifying weed patches and weed-free areas.
This has been changed
VI-based - Please elaborate at the first cite
This has been corrected.
There is no concrete conclusion at the end of the abstract section. What the study suggests for the future studies?
We have added a recommendation for future studies: “For future work on training weed prediction models, choosing training sets covering the entire sample space is recommended in favor of random sampling”. The recommendation has also been added to the overall conclusion.
Line 111. The grass weeds were A. myosuroides, please write full form at the first cite of the weed name in each section and then abbreviate the genus name
This has been corrected
Table 1. The names of the weed species should be written in full
This has been corrected
Reviewer 3 Report
In general, this paper is of interest to the agronomy science community, especially in Precision Agriculture (PA). For this aim, the use of UAVs represents an important focus for the possible practical implications. In fact, even using affordable sensors it is possible to implement operational work-flows that can be transmitted to end users. In particular, an increasing number of studies have focused on developing algorithms for the management of agricultural productions.
With this work the authors aim to identify the optimal sampling strategy for training regression models based on aerial RGB images.
The results seem to make an interesting contribution to the scientific sector, also for future application in other crops and with different field conditions. However, to extend the applicability to other crops and other areas, additional variables should be explored. Among these are the pedo-environmental and climatic characteristics, combined with the phenological and spectral (due to reflectance), of the individual plant species.
Moreover, there are some points of this manuscript that must be better written, described, clarified, justified, or can be discussed and presented more robustly.
In my opinion, this work is significant for PA and for the application of this methodology in this field. Either way, the results should be posted after the improvements mentioned above. Find more specific comments below.
Specific comments
- Abstract:
line 17: The acronym “VI-based” is not easy to understand for non-experts. It would be appropriate to specify what replaces the abbreviation “VI”. This must be done for all the acronyms present, the first time they appear in the text.
- Introduction:
line 107: for “AI” as above
- Materials and Methods:
Paragraph 2.1 “Experiments”:
There is no map of the site. It should be inserted. Orthomosaics of all sites should also be inserted.
Paragraph 2.2 “Crop and weed assessments”: Here the setting of the instrument is mainly described. Why not insert a graphical abstract?
Paragraph 2.3 “Statistical analysis”:
lines 229-239: Quoting references to the use of these statistical indicators.
line 257: Quoting reference to this index.
lines 263-274: Have you consulted works that use the same or similar methodology to evaluate the performance of the models? Please insert bibliography on this.
- Results
lines 300-305: Could the authors explain these results more clearly? What does data statistics explain? How does it relate to the results? Why, for e.g., do you have such low results for field 1?
lines 308-310: as above.
Have the authors given an explanation of why some VI (such as BLUE) give such low values of R2? Many of those shown in Tables 2 and 3 are not significant.
Figure 6: Why do the authors use five fields as training datasets and the remaining field as test data? Why was it not considered, for example, to use 2/3 as a training dataset and 1/3 as test data?
- Discussion
lines 442-443: Quoting a “milestone” would be desirable to justify this assertation.
lines443-452: Can the authors compare these results with those of previous works?
lines 491-495: Quoting some references.
line 508: Quoting other studies that use the “Youden’s index”.
line 509: What could be “other optimization criteria”? Describe.
line 514-518: Reinforce the concept by relating it to other studies.
line 545-548: Quoting other works to justify this assertation.
- Conclusion
This paragraph is poor. We should give prestige to the results obtained on the basis of comparison with other works, laying the foundations for further study.

Author Response
In general, this paper is of interest to the agronomy science community, especially in Precision Agriculture (PA). For this aim, the use of UAVs represents an important focus for the possible practical implications. In fact, even using affordable sensors it is possible to implement operational work-flows that can be transmitted to end users. In particular, an increasing number of studies have focused on developing algorithms for the management of agricultural productions.
With this work the authors aim to identify the optimal sampling strategy for training regression models based on aerial RGB images.
The results seem to make an interesting contribution to the scientific sector, also for future application in other crops and with different field conditions. However, to extend the applicability to other crops and other areas, additional variables should be explored. Among these are the pedo-environmental and climatic characteristics, combined with the phenological and spectral (due to reflectance), of the individual plant species.
Moreover, there are some points of this manuscript that must be better written, described, clarified, justified, or can be discussed and presented more robustly.
In my opinion, this work is significant for PA and for the application of this methodology in this field. Either way, the results should be posted after the improvements mentioned above. Find more specific comments below.
We have modified parts of the manuscript according to the suggestions and in particular added more descriptions and justifications in terms of more references.
We have, on purpose, not provided any detailed prediction models, as the focus of this work was on the statistical considerations. Therefore, we also believe that these considerations are applicable to a large range of crops, species and areas besides those used for illustration in the present work. However, we agree, that prediction models could gain from including additional variables as those suggested above. This has been added to the discussion.
Specific comments
- Abstract:
line 17: The acronym “VI-based” is not easy to understand for non-experts. It would be appropriate to specify what replaces the abbreviation “VI”. This must be done for all the acronyms present, the first time they appear in the text.
This has now been corrected.
- Introduction:
line 107: for “AI” as above
This has been corrected
- Materials and Methods:
Paragraph 2.1 “Experiments”:
There is no map of the site. It should be inserted. Orthomosaics of all sites should also be inserted.
Thank you for the comment. The fields are not from the same site, so it is not meaningful to show any maps. The orthomosaics from the remaining fields have been added to the supplementary material.
Paragraph 2.2 “Crop and weed assessments”: Here the setting of the instrument is mainly described. Why not insert a graphical abstract?
Thank you for the suggestion. We are, however, of the opinion that the manuscript already contains enough figures and tables.
Paragraph 2.3 “Statistical analysis”:
lines 229-239: Quoting references to the use of these statistical indicators.
The use of these statistical indicators are well known for prediction models and model selection may be found in most textbook on the subject. We prefer not to cite textbooks here.
line 257: Quoting reference to this index.
A reference has been added
lines 263-274: Have you consulted works that use the same or similar methodology to evaluate the performance of the models? Please insert bibliography on this.
A reference has been added
- Results
lines 300-305: Could the authors explain these results more clearly? What does data statistics explain? How does it relate to the results? Why, for e.g., do you have such low results for field 1?
lines 308-310: as above.
Statistics are only to be compared within fields, as e.g. will depend on the scale it is working on. This has been added to the statistics section. The RMSE in field 1 was generally low for all VIs as the weed pressure was low. This has been added to the result section.
Have the authors given an explanation of why some VI (such as BLUE) give such low values of R2? Many of those shown in Tables 2 and 3 are not significant.
Thank you for the comment. The reason for the low R2 for some fields is rather due to the low variation in the field (e.g. field 1) than to the VI (blue). This has been added to the result section. There has been no testing for significance of R2 in this work.
Figure 6: Why do the authors use five fields as training datasets and the remaining field as test data? Why was it not considered, for example, to use 2/3 as a training dataset and 1/3 as test data?
Thank you for the comment. As mentioned earlier in the manuscript, a typical goal is to make a global or joint model that is able to detect weeds in new fields not used in training the model. We chose to use five out of six fields for training the model to give largest and most diverse training set for this purpose.
- Discussion
lines 442-443: Quoting a “milestone” would be desirable to justify this assertation.
Agree. Two references have been added
lines443-452: Can the authors compare these results with those of previous works?
The results presented are not directly comparable to previous work. Within weed detection, the comparison of different sampling strategies has to our best knowledge new.
lines 491-495: Quoting some references.
A reference has been added
line 508: Quoting other studies that use the “Youden’s index”.
References have been added
line 509: What could be “other optimization criteria”? Describe.
A few examples have been added with a reference. More specifically, the area under the curve (AUC) for the receiver operating characteristic (ROC) curve, a partial ROC AUC, or sensitivity for a fixed specificity.
line 514-518: Reinforce the concept by relating it to other studies.
This part of the discussion is mainly thought of as a deeper explanation of the interpretations of the results and the choices made in the analyses. Accordingly, we do not find it relevant to relate this paragraph to other studies.
line 545-548: Quoting other works to justify this assertation.
A reference have been added
- Conclusion
This paragraph is poor. We should give prestige to the results obtained on the basis of comparison with other works, laying the foundations for further study.
The conclusion has been revised and a recommendation for future studies has been added.
This manuscript is a resubmission of an earlier submission. The following is a list of the peer review reports and author responses from that submission.
Round 1
Reviewer 1 Report
Dear authors, topic of the study is good and interesting, however, there are several concerns listed below.
1) Paper seems to use the default threshold value 0.5 of logistic regression. Generally specificity is lower than sensitivity in the paper, it means there are high chances of false positives. Why didn't the author try to optimize the threshold value of logistic regression that lowers the false positives? Why did you decide to choose the default threshold value of logistic regression?
2) One of the reasons behind low accuracy of 64-66% : authors didn't experiment with the methodologies to reduce false positives (it means saying that the model predicts species as weeds, even these are not weeds actually) .
3) How did the author choose the hyperparameters during training the models? Authors should have experimented with different hyperparameters tuning methods too. Right choice of Hyperparameters are very important for the performance of any machine learning model. It would help to reduce overfitting and increase accuracy.
4) Paper doesn't report how well the training data were fitting.(training accuracy vs epochs for different models). How did the author know that training data is not underfitting.? How did the author know training data were perfectly fit? Graphs and tables should be mentioned of accuracy and loss curves during training and testing stages both.
Author Response
Reviewer 1:
Dear authors, topic of the study is good and interesting, however, there are several concerns listed below.
1) Paper seems to use the default threshold value 0.5 of logistic regression. Generally specificity is lower than sensitivity in the paper, it means there are high chances of false positives. Why didn't the author try to optimize the threshold value of logistic regression that lowers the false positives? Why did you decide to choose the default threshold value of logistic regression?
Authors’ reply: Thank you for the comment. The default thresholds were chosen simply because sensitivity and specificity has different significances as noted be reviewer 2, so choosing an optimization criterion is difficult. This is added as a point of discussion. However, we have now changed the optimization procedure to include an optimization of the threshold by optimizing Youden’s index.
2) One of the reasons behind low accuracy of 64-66%: authors didn't experiment with the methodologies to reduce false positives (it means saying that the model predicts species as weeds, even these are not weeds actually).
Authors’ reply: The inclusion of optimization criteria mentioned above did change the accuracy, but not much for the logistic regression approach.
3) How did the author choose the hyperparameters during training the models? Authors should have experimented with different hyperparameters tuning methods too. Right choice of Hyperparameters are very important for the performance of any machine learning model. It would help to reduce overfitting and increase accuracy.
Authors’ reply: Thank you for the comment. Hyperparameters were mainly chosen as default values. We agree, that this is not optimal andthis has now been added to the discussion. However, this is a very small data set for any ML approach and optimizing too many parameters would be difficult to such a small data set, why only the variables sampled for splitting at each node was optimized by cross-validation. We forgot to add this information in the first version of the manuscript. This has now been added.
4) Paper doesn't report how well the training data were fitting.(training accuracy vs epochs for different models). How did the author know that training data is not underfitting.? How did the author know training data were perfectly fit? Graphs and tables should be mentioned of accuracy and loss curves during training and testing stages both.
Authors’ reply: Thank you for pointing this out. More information on the performance on the training data has now been added.
Reviewer 2 Report
Specific comments:
- lines 150-160: why orthomasaicking geometric performances are not shown in a table (RMSE on GCPs and other check points)? How plots boundaries are linked to imagery (by means of a GPS survey? With which instrumentation? Or simply by visual on-screen digitization?)
- lines 161-164: it's not clear how statistics are calculated from images (bare average over each plot?)
- line 237: in case of weed management, sensitivity and specificity have very different significances, therefore averaging them makes no sense;
- Figure 2: in general, the shown R2 values are poor; quite surprisingly, this happens also for ExG-ExR, which is one of the most promising indexes in detecting vegetation from RGB imagery; best performances are registered for B band (but Blue by itself, i.e. when not included at least in a band ratio, is very sensitive to haze and atmospheric conditions) and R+G+B (which is basically panchromatic): does the paper want to support the view that monochromatic or even panchromatic B&W imagery is better than RGB?
- Figure 3: no clear and quantitative information is present in the charts regarding predicted vs. measured values correlation; the many sequences of points with zero ordinate indicate a defect in the forecast model;
- Figure 6: no clear and quantitative information is present in the charts regarding predicted vs. measured values correlation;
- line 378-384: the considerations presented here throw a bad light on the actual practical usefulness of the proposed method, especially when compared with what is said at lines 405-406; the meaning should be better explained;
- also consideration in lines 387-396 should be carefully checked, because they raise doubts about the effectiveness of what has been presented.
- conclusions are pretty disorienting, especially at lines 481-485.
General comments:
- imagery is shown only in Figure 1 at a large scale; it would be interesting to see also imagery zoomed at a plot level;
- in general, the paper feeds high expectations ("This work in based on tried and tested drone RGB imagery and the well-established knowledge that clearly visible weeds with different colours compared to the background have distinct spectral signatures", "this work focused on simplicity rather than perfection") which are then betrayed by the presented results, that are generally poor, as is also evident from the considerations expressed by the authors themselves.
Author Response
Reviewer 2:
Specific comments:
- lines 150-160: why orthomasaicking geometric performances are not shown in a table (RMSE on GCPs and other check points)? How plots boundaries are linked to imagery (by means of a GPS survey? With which instrumentation? Or simply by visual on-screen digitization?)
Authors’ reply: The geometric performance of the orthomosaics is unimportant here because plots were manually cut based on visual on-screen evaluations. This has been added to the manuscript.
- lines 161-164: it's not clear how statistics are calculated from images (bare average over each plot?)
Authors’ reply: The following have been added: “The VIs were calculated for each plot based on the mean values of the R, G and B reflectance bands.”
- line 237: in case of weed management, sensitivity and specificity have very different significances, therefore averaging them makes no sense;
Authors’ reply: Thank you for the comment. We agree that for weed management, as for many other disciplines, sensitivity and specificity has very different meanings. We have added more to the discussion on this part. Balanced accuracy is mainly a tool for an overall assessment of the prediction and an easier comparison to other’s work. This has also been added.
- Figure 2: in general, the shown R2 values are poor; quite surprisingly, this happens also for ExG-ExR, which is one of the most promising indexes in detecting vegetation from RGB imagery; best performances are registered for B band (but Blue by itself, i.e. when not included at least in a band ratio, is very sensitive to haze and atmospheric conditions) and R+G+B (which is basically panchromatic): does the paper want to support the view that monochromatic or even panchromatic B&W imagery is better than RGB?
Authors’ reply: Thank you for the comment. We agree that blue by itself may be very sensitive to different atmospheric conditions, which in particular ca cause problems when monitoring fields over large spaces and in particular over time. However, for the small fields considered here the conditions does not change and is accordingly not considered a problem. Looking at the fields, it was quite obvious that the brightness of the plots was associated with the weed density. Accordingly, it made good sense that the R+G+B showed to be a promising index for some of the fields considered. In that sense, yes we support the use of panchromatic B&W imagery, but not necessarily over RGB, as other fields were better described by other indices. The focus on the present work has not been on the use of specific indices but rather on the methodology used for training on the prediction model.
- Figure 3: no clear and quantitative information is present in the charts regarding predicted vs. measured values correlation; the many sequences of points with zero ordinate indicate a defect in the forecast model;
Authors’ reply: R2 has been added to each sub-plot in figure 3. The many zeros are the plots with no weeds. We agree that the linear models are not the most appropriate models to capture the zeros due to the high variation in the vegetation index observed for the weed free plots. However, this is thought as a simple prediction model, where for practical purposes, negative predictions would be truncated at 0.
- Figure 6: no clear and quantitative information is present in the charts regarding predicted vs. measured values correlation;
Authors’ reply: We have added R2 to each sub-plot in figure 6
- line 378-384: the considerations presented here throw a bad light on the actual practical usefulness of the proposed method, especially when compared with what is said at lines 405-406; the meaning should be better explained;
Authors’ reply: Thank you for the comment. We have decided to delete the first part of the discussion (line 378-384)
- also consideration in lines 387-396 should be carefully checked, because they raise doubts about the effectiveness of what has been presented.
Authors’ reply: Thank you for the comment. We agree that this was not entirely clear and have revised the paragraph for clarification.
- conclusions are pretty disorienting, especially at lines 481-485.
Authors’ reply: The conclusion has been revised and specifically most of the lines referred to here has been deleted.
General comments:
- imagery is shown only in Figure 1 at a large scale; it would be interesting to see also imagery zoomed at a plot level;
Authors’ reply: Thank you for the suggestion. While images on plot level would be interesting to see the actual special and color distribution of the grass weeds among the crops, we have decided not to include such images here. In the present work, we only focus on the average weed density on the plot level, and only use simple color indices calculated as mean values across a plot. We do not work with the variation within the plot. On top, the work is build on drone images, and therefore no high-resolution images where it is possible to clearly distinguish weeds and crop are available.
- in general, the paper feeds high expectations ("This work in based on tried and tested drone RGB imagery and the well-established knowledge that clearly visible weeds with different colours compared to the background have distinct spectral signatures", "this work focused on simplicity rather than perfection") which are then betrayed by the presented results, that are generally poor, as is also evident from the considerations expressed by the authors themselves.
Authors’ reply: Thank you for the comment. We have revised the discussion to emphasize that the main focus of this work was not on making optimal prediction strategies, but to optimize how to select the training set, and further to consider the performance of simple strategies.